# Processing of Positive Newborn Screening Results for Congenital Hypothyroidism: A Qualitative Exploration of Current Practice in England

**DOI:** 10.3390/ijns7040064

**Published:** 2021-10-13

**Authors:** Pru Holder, Tim Cheetham, Alessandra Cocca, Holly Chinnery, Jane Chudleigh

**Affiliations:** 1Centre for Maternal and Child Health Research, School of Health Sciences, City, University of London, London EC1V 0HB, UK; Chudleigh@city.ac.uk; 2Department of Paediatric Endocrinology, Royal Victoria Infirmary, Newcastle upon Tyne Hospitals NHS Foundation Trust, Newcastle upon Tyne NE1 4LP, UK; tim.cheetham@nhs.net; 3Department of Paediatric Diabetes and Endocrinology, Evelina London Children’s Hospital, Guy’s and St Thomas’ NHS Foundation Trust, London SE1 7EH, UK; Alessandra.Cocca@gstt.nhs.uk; 4Faculty of Sports, Health and Applied Science, St Mary’s University Twickenham, Twickenham TW1 4SX, UK; holly.chinnery@stmarys.ac.uk

**Keywords:** congenital hypothyroidism, newborn bloodspot screening, communication

## Abstract

The objective of this research was to explore current communication practices for positive newborn bloodspot screening results for congenital hypothyroidism from the newborn bloodspot screening laboratory to clinicians and then families, in order to (i) understand how the pathway is implemented in practice, (ii) highlight regional differences and (iii) identify barriers and facilitators. A qualitative exploratory design was employed using semi-structured interviews across 13 newborn bloodspot screening laboratories in England. Participants included 35 clinicians and 17 NBS laboratory staff across the 13 laboratories and 18 members of relevant clinical teams. Findings illuminated variations in how positive newborn bloodspot screening results for congenital hypothyroidism are communicated in practice. This included regional variations due to historical arrangements and local resources. Contacting the appropriate person could be challenging and obtaining feedback from clinical teams to the laboratory after the child has been seen could be time consuming for those involved. Standardised communication model(s) for positive newborn bloodspot screening results for congenital hypothyroidism, which include named contact individuals, defined pathways of care and processes for feeding back to laboratories, may help to ensure the process is less labour intensive, particularly from a laboratory perspective.

## 1. Introduction

Primary congenital hypothyroidism (CHT) affects around 1 in 2000 babies in the United Kingdom every year [1] and between 1:2000 and 1:4000 worldwide [2]. CHT is a thyroid hormone deficiency syndrome resulting from inadequate production of the thyroid hormones thyroxine (T4) and tri-iodothyronine (T3) from birth. In primary CHT this occurs for two reasons: abnormal development of the thyroid gland (either absent, ectopic or hypoplastic), known as thyroid dysgenesis; and inherited enzyme deficiency, known as thyroid dyshormonogenesis [3]. Early diagnosis and treatment of babies is crucial to prevent the consequences of CHT on neurodevelopmental outcomes [4]. Newborn bloodspot screening (NBS) has been largely successful in maximising early detection of CHT so that pre-symptomatic treatment can be initiated to reduce the long-term morbidity associated with the condition if it is left untreated [5,6,7].

In the United Kingdom, NBS for CHT was initiated in 1981 to provide the earliest possible diagnosis and intervention for babies affected. Screening comprises a heel prick blood spot sample taken between days 5 and 8 of life to measure whole-blood (WB) TSH concentrations. UK national guidelines for NBS [5] define a ‘negative’ result as having a TSH concentration of < 6.0 mU/L WB (the analytical cut-off). Such results are reported as ‘CHT not suspected’. Samples with TSH greater than or equal to the analytical cut-off are retested in duplicate from the same card to generate a triplicate mean result. This provides a more definitive result. A triplicate mean TSH concentration of < 8.0 mU/L WB (the action cut-off) is classed as a ‘negative’ result and is reported as ‘CHT not suspected’ [5,8].

Positive NBS results are referred to as being ‘presumptive positive’ (PP). That is, the result is presumed to indicate a positive result until further diagnostic investigation is undertaken. A PP result is defined as a TSH concentration of > 20 mU/L and is reported as ‘CHT suspected’. Repeat samples are taken for ‘borderline’ results (concentration between 8 and 20 mU/L), for babies born at <32 weeks gestation, when there is not enough blood on the card for testing (insufficient sample) or if the card is damaged or contaminated. If the TSH level remains at 8 mU/L or above in the repeat sample, a diagnostic referral to the clinical team is required [5,8].

### Communicating Presumptive Positive NBS Results for CHT

Previous research has demonstrated that internationally, the process of communicating positive results for other conditions included in NBS varies [9,10]. Whilst the state of health of PP cases are not compromised by variation in processing (i.e., the basic end goal of neonatal screening is still met) [11], previous research has shown that variation in processing can have deleterious effects on parents, including intensified feelings of distress and anxiety [12,13,14] as well as challenging and labour-intensive processing for staff, particularly from a laboratory perspective [15]. In the UK, when a PP NBS result for CHT is identified by the laboratory, a complex communication pathway between the laboratory, clinical team and family ensues [5]. The clinical team may involve health professionals in primary, secondary or tertiary care (see Table 1 for definitions of levels of care and practitioners in the UK). A similarly complex picture is evident internationally for cystic fibrosis [16].

In the UK, national population screening programmes are implemented in the National Health Service (NHS) on the advice of Public Health England (PHE) as well as independent, evidence-based recommendations from the UK National Screening Committee (UK NSC). PHE develops standards and provides specific services that help the local NHS implement and run screening services across the country [19]. NBS Laboratory guides are available for CHT, which provides specific information on the pathway following a PP NBS result for CHT, including how suspected CHT cases should be reported and communicated from the laboratory to the clinician and from the clinician to the family [5]. Recommendations for how PP CHT cases should be reported and communicated can be seen in Table 2.

In the UK, the guidance [20] states that PP NBS results for CHT should be referred to the specialist paediatric endocrine team (regional specialist team) or to a clearly identified lead paediatrician with a special interest in CHT or experience in managing these patients. Arrangements should be in place to address issues such as managing screen positive babies around weekends and bank holidays or in the event of staff absence. Recommendations for these alternative arrangements are not specified. In addition, it is stated that this should be part of a comprehensive NBS service specification agreed with commissioners and local clinical services together with other NBS programmes [5].

In practice, the clinical referral of PP NBS results for CHT is widely variable throughout the country depending on local arrangements, resources and historical influences [15]. This is largely attributed to the fact that, whilst other screened conditions in the NBS programme (i.e., cystic fibrosis, sickle cell disease and metabolic disorders) have dedicated specialist clinical teams, CHT is managed by a mixture of general paediatricians with an interest in endocrinology as well as tertiary paediatric endocrine teams. Babies are therefore referred to paediatric teams with differing levels of expertise in CHT management; tertiary units will typically manage a newly diagnosed baby every few weeks, whilst smaller secondary care centres may manage a newly diagnosed baby with CHT every one to two years [15]. This can lead to difficulties for the laboratories in identifying who to send the clinical referral to and in terms of receiving confirmation that the child has been followed up according to national guidelines [21]. As such, there is need for a more streamlined approach to communication practices for positive NBS results for CHT; although there is no robust evidence to indicate that babies are not being followed up in a timely fashion, data from the screening programme demonstrate a wide spectrum in terms of the age at treatment initiation, and there is evidence of possible deleterious effects of poor communication practices for the family [21].

In terms of communicating a PP result to families, consensus guidelines for CHT state that these should be communicated by an experienced person (e.g., screening laboratory staff or paediatric endocrine team) either by telephone or in person [22]. However, the specific training or qualifications clinicians should have and the content of the communication is not considered in this context [23]. This is despite the fact that families often have poor pre-existing knowledge about conditions screened by NBS [24] and that, specifically for CHT, the communication of PP results can be quite complex. This is because follow-up confirmatory testing can include thyroid function tests (serum TSH and free T4) as well as ultrasonography and/or radio-isotope scanning to determine the underlying thyroid gland abnormality [5].

In practice, this can lead to inconsistency between clinicians in terms of the methods used to communicate positive NBS results and the content of the communication to parents, which can cause disparity in parents experience of receiving the NBS result [21]. Adverse experiences of receiving a PP result can have long-lasting effects for parents including subsequent relationship with healthcare professionals (HCPs), management of their child’s care and psychological impact [25]. It can also lead to difficult and stressful experiences for HCPs, particularly in situations where there are no formal mechanisms in place to support them [21]. When receiving a PP NBS result, parents value HCPs being well informed about the condition with the ability to provide additional information upon request (over and above staff familiarity) [25], the option to have both partners present at home when receiving the news [26] and the appropriate pacing of information provision so as not to overwhelm [25].

Many studies have explored communication of positive NBS results to families [9,13,14,27,28] but few have explored communication of NBS results between the laboratory and clinical teams involved in this process [21]. Indeed, the communication of positive NBS results in practice from the laboratory to families via appropriate clinicians varies widely across England, highlighting the need for a consistent, ‘best practice’ approach [15]. Despite positive NBS results for CHT being one of the most common outcomes of NBS programmes, very little research has focused specifically on the communication of positive NBS results for CHT in practice both in England and internationally. This paper expands on the work of Chudleigh et al. [21], providing a more in-depth exploration of communication practices specifically for CHT. The purpose of the current study was to explore current communication practices for positive NBS results for CHT from the NBS laboratory to clinicians and then to families to understand how the pathway is implemented in practice, highlight regional differences within England, identify barriers and facilitators and make recommendations for future practice.

## 2. Materials and Methods

A qualitative exploratory design was employed using semi-structured telephone interviews with (i) laboratory staff employed in the 13 NBS laboratories in England and (ii) members of relevant clinical teams notified of positive NBS results from the respective NBS laboratories. This study was part of a larger programme of work [29] approved by the London Stanmore ethics committee (17/LO/2102).

Setting: In England, there are 13 NBS laboratories that process the results for CHT; these comprised the study sites.

Inclusion and Exclusion Criteria: Staff employed in NBS laboratories and involved in the processing of positive NBS results for CHT and members of relevant clinical teams involved in communicating positive NBS results for CHT to parents in the previous 6 months were included. Staff who had not been involved in processing or communicating positive NBS results for CHT in the last 6 months or who had personal experience of receiving a positive NBS result were excluded.

Recruitment: A two-stage sampling approach was employed where participants were first sampled purposively based on their experience with the phenomena of interest, followed by a second stage of snowball sampling where the first participants suggested others. Directors of all 13 NBS laboratories in England were invited to participate. These were identified through the UK Newborn Screening Laboratories Network (http://www.newbornscreening.org/site/laboratory-directory.asp) [Accessed on 22 February 2018] and were contacted via email by a member of the research team. Directors of NBS laboratories were invited to be the local principal investigator for their study site and were asked to provide names and contact details of staff within the laboratory who met the inclusion criteria for the study. These staff members were contacted via email and invited to participate. Following the interview, laboratory staff were asked to identify members of local clinical endocrine teams (medical consultants, nurse specialists, specialist screening nurses) involved in communicating positive NBS results for CHT. These staff were then contacted via email and invited to participate. Written informed consent was obtained from all participants.

Data Collection: As part of a larger programme of work [29], semi-structured telephone interviews comprising closed and open-ended questions were conducted between June 2018 and February 2019. The interviews sought to identify the approaches used to communicate positive NBS results for CHT from NBS laboratories to health professionals. Data were collected on the mode of communication strategy (face-to-face, letter, telephone, e-mail), the resources involved in each communication strategy, who provides the information and their role, who arranges the initial appointment, location (co-located or alternative site) of relevant services and closure of the referral loop for each condition.

Data Analysis: The purpose of data analysis was to describe and identify current referral practices for PP CHT cases between NBS laboratories and health professionals. Qualitative data from open-ended questions were analysed using thematic analysis [30] using an inductive approach. Data from laboratory staff and clinical staff were analysed separately. Seven interview transcripts from laboratory staff were coded by two members of the research team (JC and HC) in order to aid coding comparisons and inform and align code development [31]. A code book was developed based on these jointly coded transcripts. A further seven laboratory transcripts were then coded separately by three members of the research team using the code book (JC, HC and PH). These separately coded transcripts were compared. A similar process was followed for the transcripts for clinical staff. Following this, the same members of the research team coded the remainder of the laboratory and clinical staff transcripts using the relevant code books. This was an ongoing, iterative process; new codes were developed and the definition of codes refined as the analysis progressed [32]. Once this initial coding was completed, these codes were then collapsed into themes.

## 3. Results

In total, 29 interviews were conducted; 15 interviews with 17 members of NBS laboratory staff across 13 laboratories, and 14 interviews with 18 members of clinical teams. Demographics of participants can be seen in Table 3.

Four themes were identified from the data; the first three—method of referral from laboratory to clinical team, communication of PP results from clinicians to families and arrangement of first appointment—focused on referral from the laboratory to clinicians, while the final theme focused on feedback from clinical team to laboratory. These are summarised in Figure 1 and are explored in detail below, supported using illustrative quotations from the interview data. Table 4 summarises the key sources of variation between regions, which included the individual who was notified of the positive NBS by the laboratory, the member of staff responsible for the initial contact with the family and the method used.

### 3.1. Method of Referral from NBS Laboratory to Clinical Teams

Data indicated that laboratory referrals were made to a range of different clinicians including consultants, their secretaries, paediatric junior doctors, members of the primary care team or screening coordinators. Often there was not a named individual for the laboratory to contact. As such, for many laboratories, the referral of positive NBS results for CHT was viewed as less straight forward compared with the other conditions included in the NBS programme, which often had dedicated teams to contact.


*“…congenital hypothyroidism is one of the most tricky referrals for us to do basically because we’re not phoning a team actually for that condition. …we get feedback saying, you know, ‘Why didn’t you contact the GP [Primary care practitioner]? It’s not me’… At different hospitals have slightly different ways that they want us to do it.”*
Study Site 6


*“I would love it if I just had one person to call about all my hypothyroidism babies, make my life so much easier if I didn’t have to phone different GPs [Primary care practitioners] and different consultant endocrinologists.”*
Study Site 10

Some laboratories that did have a designated consultant or specified list of clinicians to contact following a positive NBS result for CHT viewed the referral from the NBS laboratory to the clinical teams as positive.


*“… so that’s why we have to have a designated consultant. It’s a specific person who knows they’re going to do it so you never meet that barrier of, ‘Oh, I don’t want to do that. I’m not going to take that’. … So, I think because of that everybody pulls together really well.”*
Study Site 13


*“The fact that we have a bleep number for the clinical teams that we’re trying to communicate to is helpful… I think just having a tight bleep list of people that you’re communicating with is positive.”*
Study Site 5

However, some laboratories, including those that had named individuals to contact, viewed the referral of positive NBS results for CHT as often time consuming. This was due to not being able to contact the appropriate person or needing to wait for the appropriate busy clinician to return a telephone call.


*“So, whilst we have named contacts, sometimes trying to get hold of them can be difficult, particularly if, for example, contact hours have changed or people are on annual leave.”*
Study Site 4


*“With the thyroids, it can be quite difficult to get hold of the appointment time, because if we can’t get the consultant… you may be waiting for them to call you back.”*
Study Site 1


*“…it does sometimes feel like a bit of a battle trying to get hold of someone.”*
Study Site 3

Some laboratories viewed the communication process for all conditions as particularly problematic over bank holidays; although the referral to the clinical team may take place on the same working day as per the guidance, it may not be possible for the family to be seen the same or the next day after parents are informed of their baby’s positive screening result.


*“…long Bank Holiday weekends and things like that, working out how to, you know, make sure it’s processed in the correct way. … making sure we have a, sort of, set protocol for four-day weekends.”*
Study Site 12

National guidelines state that when a positive NBS result for CHT occurs, the referral from the laboratory should be made to the relevant clinicians both verbally and in writing by secure email and should include a link to the standardised diagnostic and initial treatment protocol. Even though national template letters are available from Public Health England (PHE), 10 out of 13 of the laboratories created their own templates for this purpose following further development and improvement by staff.


*“… there are pro-formas from Public Health England, but we happen to have one that we’ve been using for a long time. … Part of the problem we have with the pro-formas with Public Health England is that they’re not all in usable forms … we already had one in place that was already set up within our system that is easy for us to use.”*
Study Site 10

However, despite this, some laboratories and clinical teams recognised the need for a standardised referral template.


*“I think if there was a set standard, for each condition, if there was a standard template for this referral, that every hospital, no matter which laboratory is making the referral, they all have the same form. … so that it’s all recognised… If every laboratory produces a different referral form, then it looks slightly different.”*
Study Site 11

### 3.2. Communication of PP Results from Clinicians to Families

The national guidelines state that when a positive NBS result for CHT occurs, families should be contacted by an ‘informed health professional’. In practice, data indicated that families are contacted by a range of different clinicians including consultants, their secretaries, clinical nurse specialists, midwives, health visitors, paediatric junior doctors, primary care practitioners and screening coordinators with varying levels of experience with or knowledge of CHT. Communication of positive NBS results for CHT to families was seen as relatively straightforward to manage.


*“We do far less visits for CHT babies. They’re mainly phoned up and told about the results, and then told when the appointment is and where …It’s a lot simpler disorder.”*
Study Site 1

Clinical teams that had specialised members of staff available to deliver the results to families viewed this as positive for the families.


*“…things like screening specialist nurses would be very useful actually”*
Study Site 2


*“…if somebody else goes out… you don’t feel they’ll be about to field all the questions…it’s very much on how the person speaks to them, and what they say… we’ll always get some people who… didn’t know what it was, and didn’t know which test it was for.”*
Study Site 8

When a positive NBS result for CHT occurs, the method by which the ‘informed health professional’ communicates the result with the families differed between teams, with 8/13 centres making the first contact with parents by phone, versus 5/13 centres that initiated home visits. The choice of method depended on local resources.


*“…we do visits to thyroid babies…there’s kind of postcode lottery for that. That seems unfair.”*
Study Site 1

Following the initial phone call or home visit, a site-developed follow-up email was sent by some clinicians to families. The aim was to relay the details from the call/visit and to provide instructions for the families regarding their first clinic visit. Some clinical teams viewed the process of contacting the families as time consuming for a range of reasons including transcription errors in the contact details section of the referral form sent by the laboratories and the NBS card not being attached to the referral.


*“…we sometimes have to play a bit of detective work to get the family.”*
Study Site 1

### 3.3. Arrangement of First Appointment

When a PP result is received by the relevant member of the clinical team, a clinic appointment including confirmatory testing is arranged. The health professional responsible for arranging the appointment and diagnostic tests varies from the specialist screening coordinator, screening/specialist nurse, screening health visitor, consultant, midwife or member of the NBS laboratory. This was often centre specific and depended on local arrangements and resources. In some cases, the laboratory assumed responsibility for making the arrangements due to a lack of trust that the referral would be followed up correctly.


*“I would ask for the endocrine team to take a little bit more responsibility in the arranging an appointment… That does take up quite a bit of the time… So, there’s probably a little bit of lack of trust on my part. It is probably why I tend to take a hands-on approach… I’m quite keen to see the job through. I don’t like handing over responsibility to anybody else because, you know, there’s a life at stake.”*
Study Site 7

Some laboratories viewed the management of referrals at the clinical end, namely the timing of the first clinic appointment, as not consistent between trusts. This sometimes made it harder for the laboratories to fulfil their responsibilities.


*“So, some clinicians like to see them very promptly, some are more relaxed in the timing, still within the guidelines. So, it’s not a consistent approach, whereas the other disorders are all very clear when the clinicians actually see the children. So, it makes it easier for the nurses to go out and contact the family.”*
Study Site 1

Concerns were raised by laboratory staff and members of clinical teams about the equity of care particularly in relation to availability of scans following a positive NBS result for CHT.


*“There definitely is some variation between what tests are done, diagnostically, particularly with the congenital hypothyroidism… So, in terms of equity of care, it would seem that, given that it’s a national screening programme, people should be having the same tests for diagnosis as well.”*
Study Site 4

*“One of the problems is that the congenital hypothyroidism screening and investigation is done differently in different parts of the UK, so this is a problem. So, there are some centres that do scans, some centres that don’t do scans, so there isn’t any uniform resource.”* Study Site 9Study Site 9

### 3.4. Feedback from Clinical Teams to NBS Laboratories

In the UK, following the referral of a baby with a positive NBS result for CHT, laboratories require feedback from the relevant clinical team once the baby has been seen, assessed and confirmatory testing has been undertaken. There was no consistent unified national approach to providing this feedback, which led to time-consuming efforts by the laboratory staff to obtain information. It was considered to be particularly more challenging and time consuming for CHT compared with the other eight conditions included in the NBS programme. This was often attributed to the fact that CHT care is delivered from a number of different units within each region. Therefore, affected babies were often seen in ‘local’ centres rather than tertiary referral centres because a specialist, tertiary, trained paediatric endocrinologist was not deemed to be necessary when it came to delivering CHT care. As a result, clinicians from many more localities might be involved in their care. To remedy this, some laboratories had sought local solutions to help them deal with the difficulties associated with feedback for positive NBS results for CHT.


*“For congenital hypothyroidism… we refer to so many different consultants that it does vary between each trust. So, if we don’t receive feedback we have to phone and write letters, and that does take quite a bit of time… It’s purely because there isn’t a standardized approach as with the other conditions.”*
Study Site 1


*“CHT is much more of a problem [compared to the other eight conditions included in the NBS programme] in this region because we are not phoning one individual consultant in this region… I have to chase around a lot more to get that information from other hospitals.”*
Study Site 10

This was in contrast to the views of clinicians who were responsible for providing the feedback to the NBS laboratories who described steps they took to ensure this information was fed back to the laboratories. This suggests that there may be a discrepancy between the information the laboratories actually require, the information clinicians are providing and who is seen to have ownership of the information.


*“Then, what I will normally do then is email [the NBS laboratory] back to say, ‘Yes, the parents will be attending,’ and so on, or, if the parents declined, which has never happened, ‘Okay, they’re not coming,’ and I would assume they would follow up.”*
Study Site 7

## 4. Discussion

The purpose of the current study was to explore current communication practices for positive NBS results for CHT between the laboratory, clinical teams and families across the UK to understand how the pathway is implemented in practice, highlight regional differences, identify barriers and facilitators and make recommendations for practice. The findings of this study demonstrated clear national variation among these practices, including the method of referral from laboratory to clinical teams, methods of communication from clinicians to families, arrangement of first appointment and process of feedback from clinical teams to the laboratory. National guidelines state that the pathway for communicating positive NBS results for CHT should be part of a comprehensive NBS service specification agreed with by commissioners and local clinical services [5]. However, data from the current study indicated that this was more ad hoc and relied on local arrangements and historical agreements rather than agreed service specifications. The findings of the current study expand on those of Chudleigh et al. [15], which showed a similar national variation in models of care for the other conditions screened for in the NBS programme. The paper raises questions about the processing of positive newborn screening results for CHT, which could be addressed in future work.

CHT is generally viewed as being a disorder that can be managed appropriately by general paediatricians with an interest in CHT [5]. National guidelines state that, when a positive NBS result for CHT occurs, referrals from the laboratory should be made to either an ‘expert paediatrician’ (members of a regional specialist endocrine team and lead paediatricians with a special interest in CHT) or a local general paediatrician with support from the ‘expert paediatrician’ [5]. As such, laboratories should have a list of HCPs to contact and that there should be provisions in place for alternative arrangements should the named contact be unavailable [5]. However, the findings of this study show that in practice, some laboratories did not have a designated consultant or specified list of clinicians to contact following a positive NBS result for CHT, which made the process of making a clinical referral challenging and time consuming. It also created uncertainty among laboratory staff that the babies would be followed up appropriately. This suggests that the provision of named individuals for referring results is important in potentially alleviating communication issues between laboratories and clinical teams with regards to follow-up arrangements.

Performance data from 2017 to 2018 [11] indicated that 92.7% of babies with a CHT positive screening result had an appointment with an appropriate clinical team initiated within three working days of sample receipt by the NBS laboratory, compared to 100% of babies with MSUD, GA1, IVA and MCADD and 99.1% of babies with PKU. This may be attributable to the fact that delayed treatment of metabolic disorders is life threatening, whereas it is not for CHT [5]. In addition, 93% of children with CHT entered clinical care within the suggested condition-specific timeframe; this was higher than for those babies with MSUD (50.0%), PKU 61.1%), HCU (66.7%), CF (66.8%) and MCADD (76.2%). Therefore, while these performance data indicate that difficulties communicating the positive NBS result for CHT may have hindered timely referral to relevant clinical teams, this did not delay initiation of clinical care, which is important to avoid negative neurodevelopmental outcomes.

It is also important to note that in terms of service delivery, whilst tertiary and secondary units will provide 24 h general paediatric care, due to resource and financial limitations, the NHS does not extend to the provision of face-to-face paediatric endocrine consultations outside of out-patient clinics, which operate during standard working hours. Therefore, evidence suggests the managing team often have to decide whether babies are seen the same day by someone relatively inexperienced or the following day by a paediatric endocrinologist during standard working hours [15,21].

The study findings suggest that laboratories that did have named contacts to communicate the result to viewed the referral from the NBS laboratory to the clinical teams as positive. Indeed, for other conditions screened for in the NBS programme, which have dedicated teams to contact and close working relationships both physically and personally between laboratory staff and clinical teams, are seen to enhance the referral process [15]. This suggests that for CHT, named individuals may help to ensure the process is less labour intensive for laboratory staff. Furthermore, whilst in the present study laboratory staff recognised the importance of standardised referral templates, most laboratories used site-developed proformas. This meant that for clinical teams who received positive NBS results from more than one laboratory, or for clinical staff who moved between trusts across the UK, the information and format received from the laboratory varied. This suggests that more comprehensive, standardised templates may help clinicians who are not based in tertiary centres to understand the next steps in the process for these babies. Additionally, in the present study, the clinical referral process was seen as particularly problematic over bank holidays, where there was not always a clear protocol available, suggesting clearer guidance for bank holidays is needed for all conditions.

Geographical considerations need to be kept in mind when reflecting on the local variations described in this study and why services developed as they have. Some tertiary units are more local and/or accessible than others, with the journey from outer London to a tertiary centre likely to be more manageable for families (as an example) than in some of the larger geographical areas where travel to a tertiary unit could be more logistically challenging and/or time consuming.

In terms of communication of PP results from clinicians to families, clinicians in the present study described the process of contacting the families as time consuming for reasons including transcription errors in the contact details. This resulted in families being contacted by a range of different clinicians with varying levels of experience with or knowledge of CHT. This could be due to the fact that CHT is seen as being easier to manage and can be overseen by general paediatricians with a special interest, compared with the other conditions screened for in the NBS programme who have dedicated teams to deliver the news, and thus tend to be contacted by a smaller range of HCPs [15]. Previous research suggests that parents appreciate when the initial result is given by someone with condition-specific management knowledge and experience and would substitute staff familiarity for staff knowledge [25]. Some general paediatricians may have this expertise by virtue of their training and experience, but some may not. Similarly, in the present study, clinical teams that had specialised members of staff available to deliver the results to families, such as specialist screening nurses, viewed this as positive for the families. The importance of specialised members of staff is shown to be similarly important for the other conditions screened for in the NBS programme [21]. This suggests that standardising the content of communication to families, particularly when delivered by HCPs who have less knowledge about CHT or are less experienced in delivering the news, could improve care. Furthermore, in the present study, the method used by the informed health professional to communicate the result to the families varied from a phone call to a home visit and was dependent on local resources. This is despite the fact that communication channels are viewed as important for families, with parents of children with CHT in particular supporting telephone contact [25] and both parents being visible when receiving the news [26]. Additionally, despite previous research demonstrating that families value appropriate pacing of information provision [25] and appreciate receiving their babies’ NBS results in a written format as well as additional condition-specific information and preparation for subsequent diagnostic procedures [25]; in the present study, follow-up emails were only sent by some clinicians to families and this was not consistently done across trusts. This suggests that a standardised process of following up with families after the initial delivery of a positive CHT result may be beneficial. It also suggests that any models of care for CHT must consider these fundamental aspects: staff knowledge of CHT, communication channels, visibility of both parents (if appropriate) when receiving the news and appropriate pacing of information. Further research may be needed to find out parental preferences specifically for pacing of information.

The findings of this study demonstrate that the HCP responsible for arranging the first appointment and diagnostic tests varies. This was often centre specific and depended on local arrangements and resources. As such, laboratory staff in the present study described a lack of confidence on occasions once the referral had been made with regard to the child being followed up according to national guidelines. To alleviate this concern, findings of this study indicated that some laboratories assumed responsibility for arranging the first appointment and diagnostic tests. Clinicians in the present study also indicated that timing of the first clinic appointment was not consistent between trusts, which demonstrates a possible lack of parity in terms of provision of care for families who have received a positive NBS result for CHT. However, it is important to note that there is no evidence to indicate babies are not being followed up in a timely fashion [11].

Furthermore, guidelines reiterate the importance of ensuring that information relating to every child’s NBS journey is documented in a timely fashion [22], and in order to do so, efficient feedback from clinical teams to NBS laboratories is necessary. However, the findings of the current study suggest laboratories struggled to collate and coordinate feedback from different sources after a child had been seen and this was considered to be particularly challenging and time consuming for CHT. This was in contrast to the views of clinicians who were responsible for providing the feedback. This inconsistency suggests the need for a more unified national approach to providing feedback to laboratories.

This is the first known study to explore communication pathways for positive NBS results from the laboratory to clinical teams, specifically for CHT. Participants represented the 13 NBS laboratories in England involved in managing CHT increasing the transferability of the findings. Study design, data collection and analysis were influenced by members of the PPI advisory group and relevant charities. In terms of limitations, the researchers are experienced in this field, which may have biased data collection and analysis. This study also recognises that, due to a paucity of evidence internationally on communication pathways for positive NBS results from the laboratory to clinical teams, specifically for CHT, it may be beneficial for further exploration and evaluation to be done internationally.

The findings of this study suggest that the current model(s) for CHT provision in England would benefit from being reviewed and that different models of care for CHT provision need further exploration and evaluation. This study recognises that attempts to streamline some areas of CHT screening and management are already being made by the development of regional networks and by the formation of paediatric special interest groups. Many paediatricians recognise the need for children with CHT and indeed with many other disorders such as type 1 diabetes [33] to be managed by doctors with appropriate training and expertise. Our research suggests that a more in-depth analysis of barriers to more refined, consistent care in CHT is warranted at this juncture.

## 5. Conclusions

The findings of the present study indicate that variation in communication practices for CHT exist across the UK and are influenced by a range of factors including available resources, local arrangements and historical agreements as well as a lack of clear guidance. This has a profound influence on the methods used to communicate positive NBS results for CHT from the laboratory to clinical teams and subsequently to families as well as the content of such communication. The impact of variations in communication practices is supported by previous research which has focused on communication of carrier and affected results for other conditions included in the NBS programme both nationally [12,13,14] and internationally [16]. The findings of the present study suggest that further guidance and a more standardised pathway(s) to follow for CHT may help to ensure a more cohesive referral process that meets the needs of both parents and staff. Further exploration and evaluation work may be necessary before definitive recommendations for practice can be made. Questions that could be addressed in future work are outlined below.

### Questions That Need to Be Addressed in Future Work

What is the best way to construct communication pathways that are independent of any single person and hence will not be disrupted by staff illness, holiday or a change in personnel?What is the best way to educate clinicians managing these babies so that there is comparable care around the nation, irrespective of the nature of secondary or tertiary centre involvement?What is the best way to standardise communication with families, particularly when delivered by HCPs who have less knowledge about CHT or are less experienced in delivering the news to improve care?What is the best way to organise care pathways that take into consideration the challenges that some families will face when accessing care?To what extent are existing barriers to more refined care a reflection of resource or funding issues and to what extent do they reflect factors independent of such factors?

## Figures and Tables

**Figure 1 IJNS-07-00064-f001:**
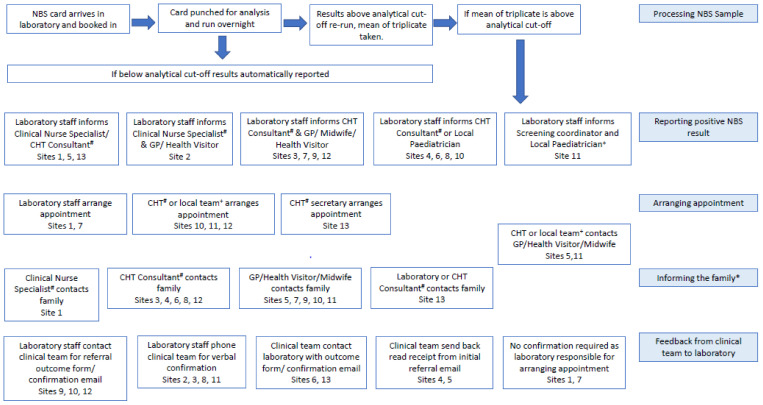
Process of positive NBS referral for CHT for each New-born Bloodspot Screening Lab. *Data unavailable for sites 2 +secondary health care team, #tertiary health care team.

**Table 1 IJNS-07-00064-t001:** Definitions of levels of care and practitioners in the UK [17,18].

Levels of Care	Definitions
Primary care	Healthcare delivered outside hospitals. It includes a range of services provided by primary care practitioners (known as general practitioners (GPs) in the UK), nurses, health visitors, midwives and other healthcare professionals as well as allied health professionals such as dentists, pharmacists and opticians. It includes community clinics, health centres and walk-in centres.
Secondary care	Secondary care is healthcare provided in local hospitals. It includes accident and emergency departments, outpatient departments, antenatal services, genitourinary medicine and sexual health clinics.
Tertiary care	Care for people needing complex treatments. People may be referred for tertiary care (for example, childhood cancer) from either primary care or secondary care.
Roles	
Clinical nurse specialist	Advanced nursing practitioners who can provide advice related to specific conditions or treatment pathways.
Consultant biochemist	Biochemists who oversee the diagnosis of disease, lead services and guide a wide range of healthcare staff.
Deputy/Director of NBS laboratory	Directors who are responsible for the overall operation and administration of the laboratory.
General paediatrician	Doctors who manage a range of medical conditions affecting children from birth to the age of 16.
Health visitor	Specialist community public health nurses, registered midwives or nurses, who specialise in working with families with a child aged 0 to 5.
Medical consultant	Senior doctors who practice in one of the medical specialties.
Midwife	Practitioners who are specially trained to deliver babies and to advise pregnant women.
Primary care practitioner	Known as ‘general practitioners’ (GPs) in the UK, who treat all common medical conditions and refer patients to hospitals and other medical services for urgent and specialist treatment.
Screening coordinator	Coordinators of screening programmes.
Screening specialist nurse	Advanced nursing practitioners who can provide advice related to specific screened conditions.
Senior/Clinical scientist	Care professionals who oversee specialist tests for diagnosing and managing disease and advise doctors on using and interpreting tests.
Registrar	Doctors in specialist training.

**Table 2 IJNS-07-00064-t002:** UK guidelines for communicating positive NBS results for CHT from laboratories to clinical teams following a presumptive positive (PP) NBS result.

Process	UK Guidelines
Method of referral from laboratory to clinical team	Verbally and in writing using available template letters by secure email including a link to the standardised diagnostic and initial treatment protocol, to either an ‘expert paediatrician’ (member of a regional specialist paediatric endocrine team/lead paediatrician with a special interest in CHT) or a general paediatrician at a local centre with support from the ‘expert paediatrician’.
Time frame for communicating PP results to the clinical team	Same or next working day of the definitive NBS result being available.
Requirements when communicating PP results to families	Laboratory, ‘expert paediatrician’ or a deputy (depending upon the agreed regional protocol) notify an ‘informed health professional’, who provides the family with the appropriate information leaflet available via the screening programme and the child’s appointment details.
Time frame for first clinic appointment	Must take place on the same day or the next day after parents are informed of their babies positive NBS result.
Arrangements and follow-up for first clinic appointment	‘Expert paediatrician’ or team managing the baby help to arrange access to diagnostic investigations and should report the outcome of the first appointment to the laboratory within 48 h. The laboratory then know that the child has entered the management pathway.

Information sourced from the NBS Laboratory guides [5].

**Table 3 IJNS-07-00064-t003:** Demographics of participants.

**NBS Laboratory Staff**		
**Profession**		**Number of Staff Interviewed**
Deputy/Director of NBS laboratory		8
Senior/Clinical Scientist		8
Consultant Biochemist		1
Length of service	Median 10.5 years	Range 1.0–22.0 years
Length of interview	Median 32.46 min	Range 16.57–47.42 min
**Clinical Teams**		
**Profession**		**Number of staff interviewed**
Medical Consultant		10
Clinical Nurse Specialist		4
Screening Specialist Nurse/Midwife		3
Screening Coordinator		1
Length of service	Median 14.0 years	Range 2.0–23.0 years
Length of interview	Median 31.92 min	Range 19.16–54.58 min

**Table 4 IJNS-07-00064-t004:** Key regional sources of variation in the processing of positive NBS referrals for CHT.

**Individuals Notified by Laboratory Team**
Family
Primary care team
Secondary health care team
Tertiary health care team
**Member of staff responsible for initial contact with family**
Clinical nurse specialist
General paediatrician (with/without specialist interest)
Health visitor
Laboratory staff
Midwife
Primary care practitioner
Registrar
Screening coordinator
Tertiary specialist
**Method of initial contact with family**
Home visit
Phone call

## Data Availability

Data are available upon reasonable request from the corresponding author subject to restrictions to preserve anonymity and personal privacy (JC). These data are not publicly available as they contain information that could compromise research participant privacy/consent. Data will be available beginning 1 year and ending 5 years after publication to researchers who propose a methodologically sound proposal. Proposals should be directed to j.chudleigh@city.ac.uk. To gain access, data requesters will need to sign a data access agreement.

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
