# Peer review of "Processing of Positive Newborn Screening Results for Congenital Hypothyroidism: A Qualitative Exploration of Current Practice in England"

_2409-515X, 2021, doi:10.3390/ijns7040064_

Round 1
Reviewer 1 Report
Major Comments:
- This paper is a valuable attempt to identify the specific problems that the UK has with the process of communicating positive screening results for CH, including feedback and time consuming. However, it contains many circumstances unique to the UK, making it difficult for researchers from other countries to be interested in it.
- Staff from various professions are involved in communicating the positive screening results to patients' families in the UK, and the unique system is complex and difficult for others to understand.
- This treatise contains the responses of the interviewees as they are and is a redundant and preliminary survey. I think it should be organized into objective information such as scoring and stated concisely.
- In the "Discussion" section, the distinction between the results obtained from this study and the authors' opinions is unclearly discussed.
Minor comments:
- L39 and L42
Is “NBS” an abbreviation for "newborn blood screening" or "newborn screening"?
- L46
Is "WB" an abbreviation for "whole blood"?
- Table. 1
The letters are too small for me to read.
- Table. 2 and Table. 3
What is the difference between "Clinical Teams" in Table 2 and "Member of staff responsible for initial contact with family" in Table 3?
- L437
Is "NHS" an abbreviation for "National Health Service"?
Reviewer 2 Report
The positive result of the screening test is undoubtedly a stressful moment for the child's parents.It is necessary to shorten the time for the definitive determination or exclusion of the diagnosis, not only due to early treatment, but also to reassure parents.The assessed work presents an analysis of this process in a decentralized system of processing positive cases of CHT screening.
I have a few remarks about the work:
1./ The work analyzes the processing of NS CHT and does not mention the state of health of detected cases of CHT, which represent the basic end goal of neonatal screening.
Despite the identified processing weaknesses, this goal can be met.
2./ In the chapter Conclusions, I would expect more specific proposals to address this issue, such as the creation of CHT regional recall centers CHT in tertiary accessible hospitals, and the like.
3./ The article is disproportionately extensive, I recommend shortening the anecdotal divisions in chapter 3.1., 3.2., 3.3.
Reviewer 3 Report
The manuscript describes the communication of congenital hypothyroidism newborn screen results from the newborn screening programs in England to the providers and families, as well as follow up of presumptive positive newborns. This is an important function of the newborn screening program and the health care system as it allows the newborns with abnormal results to be diagnosed and treated in a timely way.
It would be useful if the introduction section briefly described the organizational system of NBS in England, especially to an international audience. Who oversees NBS in the country and who makes the recommendations? As described there seems to be no cohesion between the various programs.
It would be helpful if the authors had some recommendations in the form of bullet points.
The guidelines in table 1 state time frames based on day of testing or receipt of test results rather than age of newborn. The recommendation in Europe and the US is that babies with CH should be on treatment by 2 weeks of life. Treatment should be started as soon as possible, not later than 2 weeks after birth (Trotsenburg et al., Thyroid. 2021;31;387). Guidelines based on day of testing are therefore not appropriate for the entire process which includes collecting specimens, shipping to NBS program, screening, reporting, and follow-up diagnostic testing.
Minor comments:
Line 89-90 Delete duplicate “general paediatricians”
The font in figure 1 is too small to read.
Legend of Fig 1 – Newborn Screening Lab
Line 234 – Delete duplicate “card”
Line 333 –“very” should be “vary”
The manuscripts describes mainly England but UK is also mentioned. Please clarify.
Round 2
Reviewer 1 Report
The authors have replied to the reviewer's remarks faithfully and revised the manuscript appropriately.
However, the authors should spell out some of the abbreviations in Figure 1.
1) What do the abbreviations "CNS" and "HV" in Figure 1 stand for?
Author Response
Thank you for highlighting this. CNS stands for Clinical Nurse Specialist and HV stands for Health Visitor. These abbreviations have been spelt out in Figure 1.